# Spatio-temporal distribution of foot and mouth disease outbreaks in Western Amhara region from January 2018 to June 2023

Endeshaw Demil[1], Belege Tadesse[2]*

**1** Epidemiology Department, Bahir Dar Animal Health Investigation and Diagnostic Laboratory, Bahir Dar, Ethiopia, **2** School of Veterinary Medicine, Wollo University, Dessie, Ethiopia

* tadessebelege@gmail.com

**Data Availability Statement:** All relevant data are within the paper and its Supporting Information files.

## Abstract

Foot and mouth disease (FMD) is a contagious and economically important viral disease of cloven hoofed animals caused by FMD virus. This retrospective study was conducted to determine the spatio-temporal distribution, estimate the morbidity and case fatality of FMD outbreaks in Western Amhara region of Ethiopia from January 2018 to June 2023. The FMD outbreaks reported to Bahir Dar regional laboratory and confirmed by Sandwich ELISA were used for this study. A total of 164 FMD outbreaks were reported in Western Amhara region of Ethiopia between 2018 and 2023. The highest and lowest number of FMD outbreaks were reported in 2022 (n = 42 outbreaks) and 2018 (n = 9), respectively; however, there was no statistically significant difference in the occurrence of FMD outbreaks between years (p = 0.224). There was no significant difference in case fatality and morbidity rates between years (p> 0.05). Based on months, high number of outbreaks were reported during January (n = 32) across all years and the lowest during April, June and September (n = 3 for each district). There was statistically significant difference in the occurrence of FMD outbreaks between months (p < 0.001). On average, 13.66 FMD outbreaks were reported in each month. All administrative zones in Western Amhara region reported more than one FMD outbreaks during January 2018 –June 2023. The highest and lowest outbreaks were reported from East Gojjam Zone (62 outbreaks/6 district years), and Bahirdar Special Zone and Gondar Town (2 outbreaks for each district). There was no statistically significant difference in the occurrence of FMD outbreaks between the administrative zones (p > 0.05). District wise, the highest and lowest number of FMD outbreaks were reported from Goncha Siso district (n = 14) and Banja, Simada, Zigem, Machakel, Debre Elias and Debark (with one outbreak in each district). This study indicated that FMD outbreaks regular occurs in Western Amhara region. To reduce the occurrence and spreading, animal movement and transport should be controlled, and vaccination should be implemented regularly.

## 1. Introduction

Foot and mouth disease is a contagious and economically important viral disease of cloven-footed animals [1], caused by Foot and Mouth Disease virus (FMDV) of genus Aphtovirus and

**Funding:** The author(s) received no specific funding for this work.

**Competing interests:** The authors declared no conflicts of interest.

family Picornaviridae. The disease causes vesicular lesions mainly in the oral cavity and inter-digital space [2].

Foot and mouth disease endemic in Ethiopia and outbreaks were reported every year [3, 4]. According to Jemberu et al. [3], during 2007–2012, the annual district level incidence of FMD outbreak were 0.24, 0.39 and 0.85 in the crop livestock mixed, pastoral and market-oriented districts in Ethiopia, respectively. Whereas, Aman et al. [4], reported 4.68 outbreaks per 18 district years in Amhara region. According to Tadesse et al. [5], animal level morbidity of 68.1% and 54.5% in mixed crop livestock production system and commercial dairy farms, respectively in were reported from studies in active FMD outbreaks.

In Ethiopia, different studies reported animal and herd level sero-prevalence of FMD in cattle ranging from 1.4 to 53.6% and up to 61%, respectively [6–9]. Serotype A, O, SAT 1 and SAT 2 were frequently reported as a cause of FMD outbreak in different parts of Ethiopia including Western Amhara region [3, 10–12]. The economic impact of FMD outbreaks were also reported in different parts of Ethiopia. For example, Tadesse et al. [5] reported USD 34 and USD 459.1 from crop livestock mixed and commercial dairy farms, respectively in Northwest Ethiopia. Beyi [13] also reported average mean loss of USD 108.96 from commercial dairy farms in central Ethiopia. According to Jemberu et al. [14] FMD outbreak in the crop livestock mixed system resulted in per herd mean total economic losses of USD 76.

Currently, a country works to control the occurrence of FMD in different parts of Ethiopia including Western Amhara region. Even if some zones apply ring vaccination in cattle, it was not regular and scheduled, and also did not include small ruminants. To apply effective control measures and control the spreading and occurrence of FMD, epidemiological studies to determine the spatiotemporal distribution of FMD in the region are very important. Even though few sero-prevalence studies were reported in Amhara region, there are a lack of studies on the spatio-temporal distribution of FMD, particularly in Western Amhara region after 2018. Therefore, this study was conducted to determine the spatial and temporal distribution FMD outbreaks in Western Amhara region from January 2018 to June 2023.

## 2. Materials and methods

### 2.1. Study area

The study was conducted in Western Amhara region (Fig 1). The region consists of highlands (above 2300 m above sea level (m.a.s.l)), mid highlands (1500 to 2300 masl) and lowlands (below 1500 m.a.s.l). The region is made up of nine administrative zones, namely North Gondar, South Gondar, East Gojjam, West Gojjam, Awi, West Gondar, Central Gondar, Gondar Town and Bahir Dar special zone. These administrative zones are divided into a total of 59 districts [15].

### 2.2. Source of the data

Foot and mouth disease outbreak data for the period January 2018–June 2023 were obtained from the Epidemiology Department of Bahir Dar Animal Health Investigation and Diagnostic Laboratory. The source data included information such as districts, zones, species affected, index date, number of cases, number of outbreaks, number of deaths, number of vaccine given, and number of animals at risk. For this study, an outbreak was defined as one or more ruminants showing FMD clinical signs in a district and confirmation of the outbreak by the regional laboratory. Therefore, FMD outbreak incidence was computed at the district level using the five years (January 2018–June 2023) outbreak data.

Verbal consent was made with Epidemiology Department of Bahir Dar Animal Health Investigation and Diagnostic Laboratory to use the retrospective data only for this study. Since

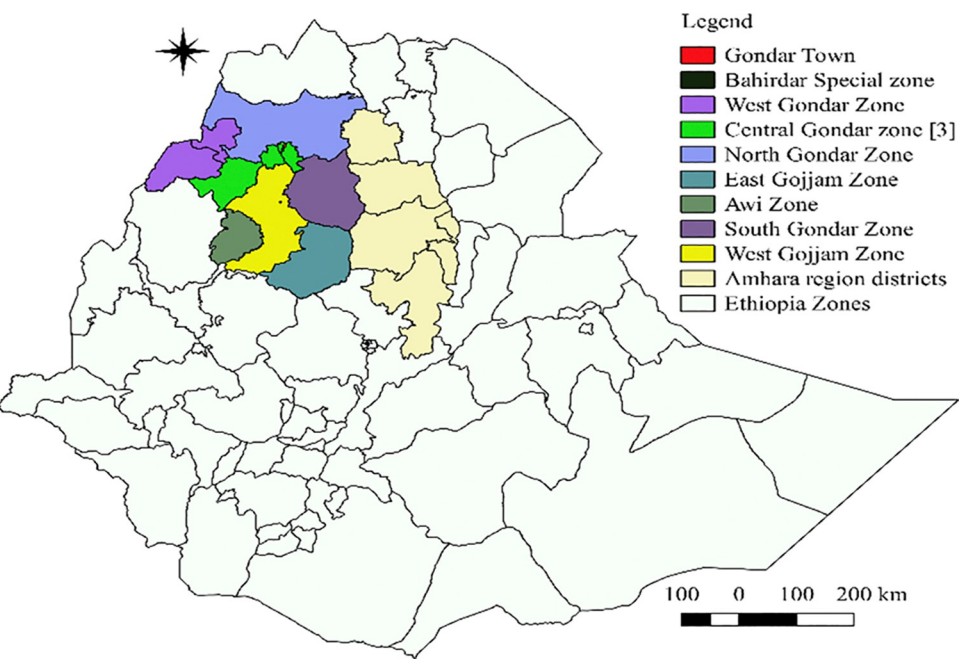

**Fig 1. Maps of the study area drawn by using QGIS 2.3.**

the study did not involve either human or animal participants or experiments (the data relays only on previously collected data), an approval from an institutional review board (ethics committee) was not needed.

## 2.3. Data analysis

Microsoft Excel spreadsheets (Microsoft Corporation) were used to manage the data and draw graphs. Descriptive methods were used to calculate outbreak incidence. The mean FMD outbreak incidence was calculated by summing all reported FMD outbreaks over the study period in Western Amhara region and divided by the total number of districts and number of years (district years). The spatial distribution of FMD outbreaks over the study period was drawn by administrative zones and districts using QGIS version 2.3. The number of FMD outbreaks reported in the five-year period was graphed to visualize the temporal trends of the disease. The graph was inspected for the presence of seasonality.

## 2.4. Study design

A retrospective study was conducted to assess the spatio-temporal distribution of FMD outbreaks in Western Amhara region.

## 3. Result

### 3.1. Temporal distribution of FMD outbreaks in Western Amhara region

From January 2018 to June 2023, 164 clinically diagnosed and laboratory confirmed FMD outbreaks were reported to Epidemiology Department of Bahir Dar Animal Health Investigation and Diagnostic Laboratory from different districts in Western Amhara region. During January 2018 to June 2023, high number of FMD outbreaks were reported in 2022 (n = 42 outbreaks) and 2021 (n = 37), while the lowest number of outbreaks were reported in 2018 (n = 9).

**Table 1. The number of FMD outbreaks, number of cases and case fatality rate during January 2018 to June 2023.**

| Year | Number of FMD cases | Number of deaths due to FMD | Population at risk | Morbidity rate | Case fatality rate | Number of outbreaks |
|------|---------------------|------------------------------|--------------------|----------------|---------------------|---------------------|
| **2018** | 591 | 5 | 48716 | 1.21 | 0.85 | 9 |
| **2019** | 1343 | 22 | 135577 | 0.99 | 1.64 | 21 |
| **2020** | 2314 | 25 | 136096 | 1.70 | 1.084 | 29 |
| **2021** | 1102 | 28 | 148089 | 0.74 | 2.54 | 37 |
| **2022** | 6948 | 35 | 261857 | 2.65 | 0.50 | 42 |
| **2023** | 7026 | 82 | 282277 | 2.49 | 1.17 | 26 |
| **Overall** | 19333 | 199 | 1012612 | 1.91 | 1.02 | 164 |
| | $X^2 = 38$; p value = 0.234 | $X^2 = 30$; p value = 0.224 | | $X^2 = 34$; p value = 0244 | $X^2 = 30$; p value = 0.224 | $X^2 = 48$; p value = 0.243 |

Note: $X^2$ = Chi-square

Although there was a variation in the number of outbreaks, there was no significant difference in the occurrence of FMD outbreaks between the years (p > 0.224) (Table 1). The case fatality rate was high (2.54) during 2021, while it was lowest during 2022 (0.50) (Table 1).

High number of outbreaks were reported during the dry seasons from December to March compared to other seasons. The highest number of outbreaks were reported in the month of January (n = 32) across all years and the lowest in April, June and September (n = 3 for each district). There was a significant difference in the occurrence of FMD outbreaks between months (p < 0.001) (Table 2). In general, the number of FMD outbreaks were above average for the months December to March and below average for all other months (Table 2). On average, 13.66 FMD outbreaks were reported in each month from January 2018 to June 2023.

## 3.2. Spatial distribution of FMD disease outbreaks in Western Amhara region

During the period from January 2018 –June 2023, FMD outbreak has been reported from all administrative zones (n = 9) of Western Amhara region. All administrative zones in Western Amhara region reported more than one FMD outbreaks during the period. A total of 164 FMD outbreaks were reported during the study period. Most of these outbreaks were from

**Table 2. The monthly number of FMD outbreaks, number of cases and case fatality rate during January 2018 to June 2023.**

| Months | Number of FMD cases | Number of deaths due to FMD | Morbidity rate | Case fatality rate | Number of outbreaks |
|--------|---------------------|------------------------------|----------------|---------------------|---------------------|
| **January** | 3909 | 19 | 1.73 | 0.48 | 32 |
| **February** | 3786 | 18 | 2.03 | 0.47 | 29 |
| **March** | 5973 | 89 | 3.86 | 1.49 | 30 |
| **April** | 345 | 0 | 0.93 | 0 | 3 |
| **May** | 586 | 0 | 1.26 | 0 | 6 |
| **June** | 469 | 0 | 0.99 | 0 | 3 |
| **July** | 180 | 4 | 1.24 | 2.22 | 4 |
| **August** | 589 | 11 | 1.41 | 1.87 | 11 |
| **September** | 770 | 12 | 1.61 | 1.56 | 3 |
| **October** | 1162 | 18 | 3.30 | 1.55 | 7 |
| **November** | 294 | 6 | 1.51 | 2.04 | 8 |
| **December** | 1269 | 22 | 0.81 | 1.73 | 28 |
| **Overall** | 19333 | 199 | 12.38 | 1.03 | 164 |
| | $X^2 = 132$; p value = 0.233 | $X^2 = 96$; p value = 0.263 | $X^2 = 132$; p value = 0.233 | $X^2 = 108$; p value = 0.252 | $X^2 = 78.5$; p value = 0.001 |

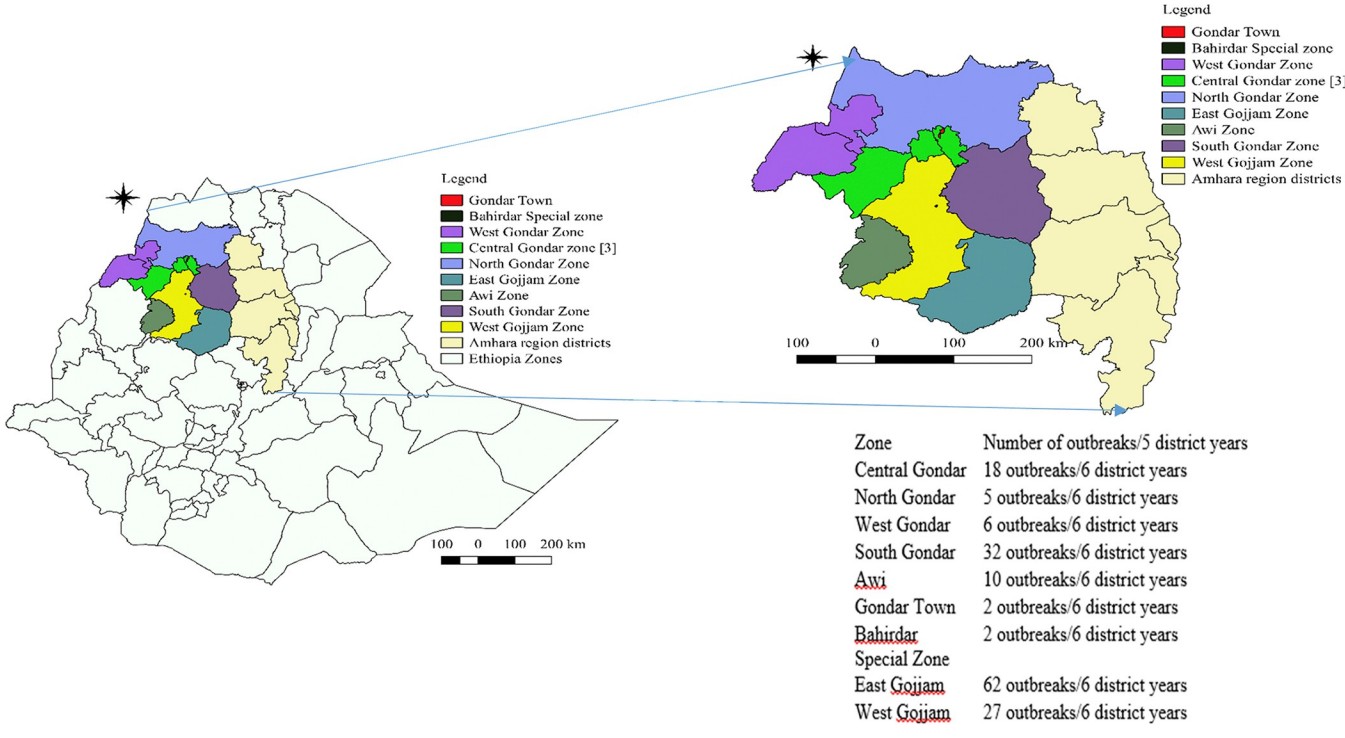

**Fig 2. Distribution of FMD outbreaks in zones of Eastern Amhara region during January 2018 to June 2023.**

East Gojjam Zone (62 outbreaks/6 district years); South Gondar Zone (32 outbreaks/6 district years) and West Gijjam (27 outbreaks/6 district years). While the lowest incidences were observed in Bahir Dar Special Zone and Gondar Town (2 outbreaks for each district). The incidence of the FMD outbreak in Western Amhara Zones per six district years is depicted in (Fig 2). The FMD outbreak incidence was above the average incidence of the region in Central Gondar, South Gondar, East and West Gojjam (Fig 2). There was no significant difference between all administrative zones of the study area ($X^2 = 63$; $p = 0.243$) in the occurrence of FMD outbreaks (Fig 2). District wise, the high number of FMD outbreak was reported from Goncha Siso district ($n = 14$) followed by Dera district ($n = 8$); while it was low in Banja, Simada, Zigem, Machakel, Debre Elias and Debark, with one outbreak in each district (Fig 3) and the difference was significant ($X^2 = 78.1$; $p = 0.002$).

### 3.3. Serotypes identified from the outbreaks

Serotype SAT2 was identified from all zones, serotype O, A and SAT1 were also identified from majority of the zones (Table 3).

## 4. Discussion

This study indicated that FMD outbreaks occurred in Western Amhara region every year with a total of 164 outbreaks in the last six years and on average 16.58 FMD outbreaks annually. This finding suggested that FMD is endemic in Western Amhara region and agrees with the report of Aman et al. [4], who reported that FMD was endemic in the region. However, in number of outbreaks, this finding was lower as compared to the 636 FMD outbreaks reported in the whole Amhara region [4].

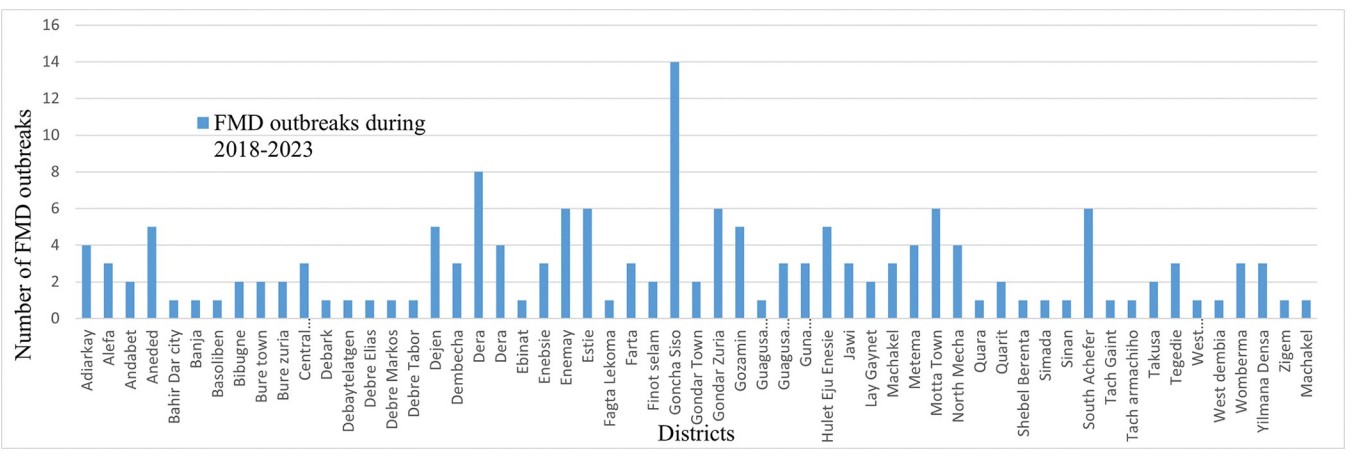

**Fig 3. Distribution of FMD outbreaks in districts of Eastern Amhara region during January 2018 to June 2023.**

The outbreaks were caused by serotype A, O, SAT1 AND SAT2. SAT 2 was identified from all zones, serotype O were also identified from a majority of the zones. This finding was in agreement with the report of Tadesse et al. [10], who reported that serotype O was identified from South Gondar, Central Gondar and Gondar Town. Serotype O, A and SA2 are the commonly isolated serotypes in Amhara region including Western Amhara region [3, 16]. Even if the vaccine includes serotype A, O and SAT2, due to irregular and lack of mass vaccination still the FMD outbreak occurred regularly in Western Amhara region.

In this study, the highest number of outbreaks were reported during the dry season in January (n = 32) and the lowest in rainy season such as June and September. This is in agreement with the report of Aman et al. [4] and Nejash [17], who reported that outbreaks FMD were higher during dry seasons. In this study averagely, 13.66 FMD outbreaks were reported in each month and the outbreaks increase year to year, which indicates that FMD outbreak frequently occurs in the Western Amhara region. Similar pattern of FMD outbreak occurrence was also reported in Sri Lanka [2]. However, Jemberu et al. [3], reported that FMD occurrence was not affected by season. The variation of FMD outbreak in different season would be due to the high movement and contact of animal movements in the dry season in the search of feed and water than rainy season and also increased transport of animals to different markets in the dry season [18].

The average morbidity and case fatality rates of FMD outbreaks in this study were 1.91% (95% Confidence interval (CL), 0.74–2.65) and 1.02 (95% CL, 0.5–2.54), respectively. This finding was lowered as compared to the 68.1% morbidity and 0.4% case fatality rate reported from study in active outbreak in Northeast Ethiopia [5]. The variation might be due to the difference in the data source, population at risk, study period and type of study. The morbidity rate in the current study is also lower than the finding of [14], who reported 2.4% morbidity in

**Table 3. Serotypes identified from outbreaks of the different zones included in the study.**

| Zones | Identified Serotypes of FMD virus | Zones | Identified Serotypes of FMD virus |
|---|---|---|---|
| **Gondar Town** | A, O, SAT1, SAT2 | West Gondar | SAT2 |
| **West Gojjam** | A, O, SAT1, SAT2 | North Gondar | SAT1, SAT2 |
| **East Gojjam** | A, O, SAT1, SAT2 | Awi | SAT1, SAT2 |
| **South Gondar** | SAT2, A, O, | Central Gondar | A, O, SAT1, SAT2 |
| **Bahirdar Special Zone** | SAT2 | | |

North Wollo, whereas the case fatality was higher than the 0.12% case fatality rate reported by Negussie et al. [16] in Ethiopia.

Foot and mouth disease outbreaks were reported from all zones of Western Amhara region during January 2018 to June 2023, with the highest of 62 outbreaks in East Gojjam. This finding was lowered as compare to the report of Ayelet et al. [19] and Jemberu et al. [3], who reported the highest outbreak in North Shewa zones of Amhara region. The high incidence of FMD outbreak in East Gojjam than other zones in the Western Amhara region might be due to high animal movement and transport in the zone between markets. Awel et al. [20], explained that animal movement and mixing of different herds exacerbates the distribution and occurrence of FMD outbreaks. Auty et al. [21] also discussed that the management system, season of the year, species of animals, and animal contact in market, grazing and watering areas results in the increase of occurrence of foot and mouth disease outbreaks.

The current study recognized that majority of the districts in Western Amhara region reported at least one FMD outbreak in six years. This is comparable with the report of Aman et al [4] and Jemberu et al. [3], who reported more than 50% of the Districts in Amhara region were affected by FMD outbreaks. This indicated that FMD was endemic and frequently occurring in Amhara region including Western Amhara region. According to the findings in this study, FMD occurs irregular (sporadically) at district level, which is in agreement with Belayneh et al. [22] and Aman et al. [4] report. The difference in the occurrence of FMD between districts might be due to the variation in the animal movement, availability of geographical barriers like mountain and river, and level of vaccination. According to Shurbe et al. [18] and Dubie and Negash [23], the occurrence and distribution of FMD outbreaks are influenced by production system, geographic location, species, and age of animals, season of the year, animal mixing, species, and breed of animals.

Overall, in Western Amhara region, the number of FMD outbreaks increases year to year during the study period. Belayneh et al. [22], also reported that increased number of FMD outbreak occurred in Amhara region. The increase in the occurrence of outbreaks could be due to transmission of the foot and mouth disease virus via air in addition to the frequent contact of different herds [24]. The current study was conducted based on the FMD outbreaks reported to Bahir Dar regional veterinary laboratory from Western Amhara region. Some districts may not report all outbreaks occurred in the districts especially if the outbreak was very mild. This may underestimate the number of outbreaks occurring during the study period. The authors are not also sure whether all the samples for confirmation of the outbreak were sampled during three to four days of the outbreak. Even if this study might have some limitations, it provides important information regarding the spatio-temporal distribution, morbidity and case fatality rate of FMD outbreaks in Western Amhara region, which could be an important input to design effective control measures against the disease.

## 5. Conclusion and recommendation

Foot and mouth disease outbreak was reported in all zones of Western Amhara region with different morbidity and case fatality rates. Serotype O, A, SAT1 and SAT2 were the cause of the outbreaks. The highest number of outbreaks and high case fatality rate was reported in East Gojjam Zone; while the lowest outbreaks were reported in Bahir Dar special zone and Gondar Town. In relation to season the highest and lowest number of outbreaks were reported during March and (September, June and April), respectively. The highest and lowest number of FMD outbreaks were reported in 2022 and 2018, respectively. Highest number of FMD outbreak was reported from Goncha Siso district (n = 14). This findings suggest that FMD outbreaks frequently occur in Western Amhara region and needs the application of effective

control measures. Therefore, to limit spreading of the disease and reduce its occurrence mass vaccination should be regularly applied, movement of animals, and mixing of different herds should be limited and awareness should be created to the livestock owners about the transmission and control mechanisms.

## Supporting information

**S1 Data. FMD data.**
(XLSX)

## Acknowledgments

We are thankful to Epidemiology Department, Bahir Dar Animal Health Investigation and Diagnostic Laboratory for their willingness to provide the FMD outbreak data for this study.

## Author Contributions

**Conceptualization:** Belege Tadesse.

**Data curation:** Endeshaw Demil.

**Formal analysis:** Endeshaw Demil, Belege Tadesse.

**Investigation:** Endeshaw Demil.

**Validation:** Endeshaw Demil, Belege Tadesse.

**Writing – original draft:** Endeshaw Demil, Belege Tadesse.

**Writing – review & editing:** Endeshaw Demil, Belege Tadesse.

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
