## [Decision Letter · Decision Letter 0]

24 Sep 2024

PONE-D-24-26790Spatio-Temporal Distribution of Foot and Mouth Disease Outbreak in Western Amhara Region from January 2018 to June 2023PLOS ONE

Dear Dr. Tadesse,

Thank you for submitting your manuscript to PLOS ONE. After careful consideration, we feel that it has merit but does not fully meet PLOS ONE’s publication criteria as it currently stands. Therefore, we invite you to submit a revised version of the manuscript that addresses the points raised during the review process.

**ACADEMIC EDITOR:** Please review the English language thoroughly. The reviewers comments are attached below, Please revise the manuscript, addressing all points raised by the reviewers. ==============================

We look forward to receiving your revised manuscript.

Kind regards,

Nussieba A. Osman, Dr. Med. Vet.

Academic Editor

PLOS ONE

Journal requirements: 1. When submitting your revision, we need you to address these additional requirements. Please ensure that your manuscript meets PLOS ONE's style requirements, including those for file naming. The PLOS ONE style templates can be found at https://journals.plos.org/plosone/s/file?id=wjVg/PLOSOne_formatting_sample_main_body.pdf and https://journals.plos.org/plosone/s/file?id=ba62/PLOSOne_formatting_sample_title_authors_affiliations.pdf. 2. In the ethics statement in the Methods, you have specified that verbal consent was obtained. Please provide additional details regarding how this consent was documented and witnessed, and state whether this was approved by the IRB. 3. In the online submission form, you indicated that [The data underlying the results presented in the study are available from the corresponding author based on reasonable request]. All PLOS journals now require all data underlying the findings described in their manuscript to be freely available to other researchers, either 1. In a public repository, 2. Within the manuscript itself, or 3. Uploaded as supplementary information.This policy applies to all data except where public deposition would breach compliance with the protocol approved by your research ethics board. If your data cannot be made publicly available for ethical or legal reasons (e.g., public availability would compromise patient privacy), please explain your reasons on resubmission and your exemption request will be escalated for approval. 

Reviewers' comments:

Reviewer's Responses to Questions

**Comments to the Author**

1. Is the manuscript technically sound, and do the data support the conclusions?

Reviewer #1: Partly

Reviewer #2: Partly

2. Has the statistical analysis been performed appropriately and rigorously? 

Reviewer #1: Yes

Reviewer #2: N/A

3. Have the authors made all data underlying the findings in their manuscript fully available?

Reviewer #1: Yes

Reviewer #2: Yes

4. Is the manuscript presented in an intelligible fashion and written in standard English?

Reviewer #1: Yes

Reviewer #2: No

5. Review Comments to the Author

Reviewer #1: Review Comments to the Author

The objective of the study is clear objective. It aims to determine the spatio-temporal distribution and estimate the morbidity and case fatality of foot-and-mouth disease (FMD) outbreaks in the Western Amhara region of Ethiopia.

The study is well defined as the authors are carry out the period, from January 2018 to June 2023, offering a perspective over 5.5 years.

The methodology is fixed as the use of data confirmed by sandwich ELISA from the laboratory.

The statistical tests were performed to compare differences between years, months, and administrative zones.

Presentation of results by year, month, and administrative zone, with specific figures are also detailed.

Conclusion and recommendations: The abstract concludes with suggestions to reduce the occurrence and spread of the disease.

For improvement

It would be useful to include a brief comparison with other regions of Ethiopia or neighboring countries to better contextualize the results.

Include an analysis of potential factors influencing outbreaks (climate, farming practices, animal movements, etc.).

Economic implications; Add an estimate of the economic impact of these outbreaks to emphasize the importance of the study.

More detailed recommendations: Elaborate further on control and prevention strategies, including details on recommended vaccination programs.

Reviewer #2: The manuscript entitled “Spatio-temporal distribution of Foot and Mouth Disease Outbreak in Western Amhara Region from January 2018 to June 2023” aimed to study the spatial and temporal characteristics of FMD outbreaks. The topic is interesting to epidemiologists, researchers and policymakers but there are some concerns in the methodology and writing that need to be addressed.

1. Introduction:

The authors have mentioned the necessity of having spatio-temporal studies for disease control in the country. However, there is limited justification on why the disease is so important in Ethiopia. It would be more convincing to show some data on impact of the disease for eg. Economic impact of FMD.

In the last paragraph, the author has shown that there are limited studies on the spatio-temporal distribution of FMD in Eastern Amhara region after 2018, but then they decided to conduct study in Western Amhara region. Do they mean Western Amhara region, as mentioned in the title? It is confusing.

Also, in the introduction part, there is no information on the measures applied in Ethiopia for the control of FMD. What kind of measures are applied for the control of the disease in Western Amhara? Is there any vaccination program against FMD in the country? What kind of animals are focused for vaccination?

2. Materials and Methods:

The authors have mentioned that the study was conducted in Western Amhara region which is made up of nine administrative zones. However, it is not the same in the given Figure 1. The demarcation of regions and zones of Ethiopia are confusing in the given map.

The authors have not stated what is their study population and susceptible species present in the study area.

In the data analysis, the authors have mentioned the Eastern Amhara. It is confusing. There is no any graph showing the seasonality.

3. Result:

Were all the 164 cases laboratory confirmed? Is it Eastern Amhara region or Western again?

Figure 2 caption again has Eastern Amhara. From Figure 2, it is not possible to know which region has how many outbreaks in a particular year. The spatio-temporal disease occurrence cannot be depicted from the figure. It is recommended to use separate figures with color gradient for different years.

Figure 3 description mentions that high number of FMD outbreak was reported from Goncha Sise district. However, in the figure, there is no mention of Goncha Sise district. It is better to show the p-value in the description, or in the table.

Also, it is recommended to graphically present the outbreak data according to year and month. This could provide more clarity.

Table 3 does not have caption.

4. Discussion:

Authors need to check the study area in all over the manuscript.

Authors need to discuss on why there is high incidence of FMD in Eastern Amhara region/Western Amhara region?

5. What does the range in the bracket mean? Is it Confidence Interval Range? If so, authors should mention at what level of CI?

Also discuss about the control measures applied in the country. If vaccination is used, what serotypes of vaccines are used?

6. Recommendation:

Do authors think disease surveillance could be important in control of FMD in Ethiopia?

Other comments:

As the outbreaks were confirmed only by ELISA, are the authors confident that all the samples were collected within 3-4 days of outbreak. It is because, ELISA can detect the antigen during the earlier period of the disease infection. For the later stages of sample collection, PCR is generally used. The authors can also add this in their study limitation.

Lastly, I suggest the authors carefully revise the manuscript, particularly focusing on grammar, spelling and overall clarity.

6. PLOS authors have the option to publish the peer review history of their article (what does this mean?). If published, this will include your full peer review and any attached files.

Reviewer #1: **Yes: **Ratiba BAAZIZI

Reviewer #2: **Yes: **Sujeeta Pokharel Dhakal

---

## [Author Response · Author response to Decision Letter 0]

7 Oct 2024

Response to reviewer 1

The authors have a great thank for the authors for their scientific comments and suggestions that are very important for the improvement of the paper. The manuscript was revised based on the comments and the response for each comment are included below. 

Reviewer #1: Review Comments to the Author

The objective of the study is clear objective. It aims to determine the spatio-temporal distribution and estimate the morbidity and case fatality of foot-and-mouth disease (FMD) outbreaks in the Western Amhara region of Ethiopia.

The study is well defined as the authors are carry out the period, from January 2018 to June 2023, offering a perspective over 5.5 years.

The methodology is fixed as the use of data confirmed by sandwich ELISA from the laboratory.

The statistical tests were performed to compare differences between years, months, and administrative zones.

Presentation of results by year, month, and administrative zone, with specific figures are also detailed.

Conclusion and recommendations: The abstract concludes with suggestions to reduce the occurrence and spread of the disease.

For improvement

It would be useful to include a brief comparison with other regions of Ethiopia or neighboring countries to better contextualize the results.

Response: Comparison of the finding in this research was made with other reports in different part of the country in the discussion section.

Include an analysis of potential factors influencing outbreaks (climate, farming practices, animal movements, etc.).

Response: Explanation of the impact of different risk factors on the occurrence of FMD outbreaks were added in the discussion section.

Economic implications; Add an estimate of the economic impact of these outbreaks to emphasize the importance of the study.

Response: Even if the reviewer suggested to estimate the economic impact of the outbreaks included in this study, in this revision we have not address the concern. Because the study was based on retrospective data from ceased outbreaks, and even if the outbreaks have huge impacts on the income of farmers, the authors couldn’t estimate the exact economic losses.

More detailed recommendations: Elaborate further on control and prevention strategies, including details on recommended vaccination programs.

Response: Additional recommendations were included (mass vaccination and limiting mixing of herds). 

Response to Reviewer #2: Comments

The authors have a great thank for the authors for their scientific comments and suggestions that are very important for the improvement of the paper. The manuscript was revised based on the comments and the response for each comment are included below. 

The manuscript entitled “Spatio-temporal distribution of Foot and Mouth Disease Outbreak in Western Amhara Region from January 2018 to June 2023” aimed to study the spatial and temporal characteristics of FMD outbreaks. The topic is interesting to epidemiologists, researchers and policymakers but there are some concerns in the methodology and writing that need to be addressed.

1. Introduction:

The authors have mentioned the necessity of having spatio-temporal studies for disease control in the country. However, there is limited justification on why the disease is so important in Ethiopia. It would be more convincing to show some data on impact of the disease for eg. Economic impact of FMD.

Response: Some justification on the economic impact of the diseases was included in this revision. Line 53-58.

In the last paragraph, the author has shown that there are limited studies on the spatio-temporal distribution of FMD in Eastern Amhara region after 2018, but then they decided to conduct study in Western Amhara region. Do they mean Western Amhara region, as mentioned in the title? It is confusing.

Response: It is editorial mistake, it is to mean Western Amhara region and it was corrected accordingly.

Also, in the introduction part, there is no information on the measures applied in Ethiopia for the control of FMD. What kind of measures are applied for the control of the disease in Western Amhara? Is there any vaccination program against FMD in the country? What kind of animals are focused for vaccination?

Response: In Western Amhara region, there are not regular vaccination programs, and only some zones apply ring vaccination in cattle. Sheep and goats are not included in the vaccination program.

2. Materials and Methods: 

The authors have mentioned that the study was conducted in Western Amhara region which is made up of nine administrative zones. However, it is not the same in the given Figure 1. The demarcation of regions and zones of Ethiopia are confusing in the given map.

The authors have not stated what is their study population and susceptible species present in the study area.

Response: The map was modified. The study population were cattle population in the study areas. Even if sheep and goats were susceptible

In the data analysis, the authors have mentioned the Eastern Amhara. It is confusing. There is no any graph showing the seasonality.

Response: It is editorial error. It was corrected as Western Amhara region.

The seasonality of the outbreak could be observed from table 2 that showed the monthly distribution of outbreaks

3. Result:

Were all the 164 cases laboratory confirmed? Is it Eastern Amhara region or Western again?

Response: Yes all the 164 outbreaks were confirmed by laboratory. In Western Amhara region.

Figure 2 caption again has Eastern Amhara. From Figure 2, it is not possible to know which region has how many outbreaks in a particular year. The spatio-temporal disease occurrence cannot be depicted from the figure. It is recommended to use separate figures with color gradient for different years.

Response: Sorry for the repeated problem, it was to say Western Amhara region. The temporal distribution (distribution of FMD outbreaks during the study period or five years period) was shown in table one. Whereas, Figure 2 showed only the overall zonal distribution of FMD outbreaks (i.e. zonal spatial distribution). 

Figure 3 description mentions that high number of FMD outbreak was reported from Goncha Sise district. However, in the figure, there is no mention of Goncha Sise district. It is better to show the p-value in the description, or in the table.

Response: Thank you for the comment. It was editorial problem it is to mean Goncha Siso district. The p-value of 3 was indicated in the description of figure 3.

Also, it is recommended to graphically present the outbreak data according to year and month. This could provide more clarity.

Response: Even if the reviewer suggest to present the yearly and monthly data graphically, it was not made because already these data were presented in table. We have been selected table to present table because, using a table it is possible to show the number of outbreaks, morbidity, case fatality, number of population at risk during each month and year clearly as compared to graphs.

Table 3 does not have caption.

Response: Caption for table 3 was included.

4. Discussion:

Authors need to check the study area in all over the manuscript.

Authors need to discuss on why there is high incidence of FMD in Eastern Amhara region/Western Amhara region?

Response: The study area was corrected in all areas as Western Amhara region. The possible reasons for the high number of FMD outbreaks were discussed in the discussion section.

5. What does the range in the bracket mean? Is it Confidence Interval Range? If so, authors should mention at what level of CI?

Also discuss about the control measures applied in the country. If vaccination is used, what serotypes of vaccines are used?

The values in the bracket in line 189-190 indicates 95% confidence interval and corrected accordingly. The control measures implemented in the country were discussed. The serotypes included in the vaccine were serotype A, O and SAT2. This was indicated in the discussion (line 178-80).

6. Recommendation:

Do authors think disease surveillance could be important in control of FMD in Ethiopia?

Response: Sure surveillance is very important for the control of FMD in Ethiopia. Because the status of the disease and any precipitating factors can be identified using a surveillance and based on the surveillance data, interventions can be applied.

Other comments:

As the outbreaks were confirmed only by ELISA, are the authors confident that all the samples were collected within 3-4 days of outbreak. It is because, ELISA can detect the antigen during the earlier period of the disease infection. For the later stages of sample collection, PCR is generally used. The authors can also add this in their study limitation.

Response: Thanks for the comment. We are not sure that all samples were collected within 3-4 days of the outbreak. This was added as a limitation of the study. Line 227-229.

Lastly, I suggest the authors carefully revise the manuscript, particularly focusing on grammar, spelling and overall clarity.

Response: The whole manuscript was revised and edited.

---

## [Decision Letter · Decision Letter 1]

23 Oct 2024

Spatio-Temporal Distribution of Foot and Mouth Disease Outbreaks in Western Amhara Region from January 2018 to June 2023

PONE-D-24-26790R1

Dear Dr. Tadesse,

We’re pleased to inform you that your manuscript has been judged scientifically suitable for publication and will be formally accepted for publication once it meets all outstanding technical requirements.

Kind regards,

Nussieba A. Osman, Dr. Med. Vet.

Academic Editor

PLOS ONE

Reviewers' comments:

Reviewer's Responses to Questions

**Comments to the Author**

1. If the authors have adequately addressed your comments raised in a previous round of review and you feel that this manuscript is now acceptable for publication, you may indicate that here to bypass the “Comments to the Author” section, enter your conflict of interest statement in the “Confidential to Editor” section, and submit your "Accept" recommendation.

Reviewer #2: All comments have been addressed

2. Is the manuscript technically sound, and do the data support the conclusions?

Reviewer #2: Partly

3. Has the statistical analysis been performed appropriately and rigorously? 

Reviewer #2: Yes

4. Have the authors made all data underlying the findings in their manuscript fully available?

Reviewer #2: Yes

5. Is the manuscript presented in an intelligible fashion and written in standard English?

Reviewer #2: Yes

6. Review Comments to the Author

Reviewer #2: (No Response)

7. PLOS authors have the option to publish the peer review history of their article (what does this mean?). If published, this will include your full peer review and any attached files.

Reviewer #2: **Yes: **Sujeeta Pokharel Dhakal

---

## [Editor Report · Acceptance letter]

19 Nov 2024

PONE-D-24-26790R1 

PLOS ONE

Dear Dr. Tadesse, 

I'm pleased to inform you that your manuscript has been deemed suitable for publication in PLOS ONE. Congratulations! Your manuscript is now being handed over to our production team.

Kind regards, 

on behalf of

Dr. Nussieba A. Osman 

Academic Editor

PLOS ONE